# Chloroform-Injection (CI) and Spontaneous-Phase-Transition (SPT) Are Novel Methods, Simplifying the Fabrication of Liposomes with Versatile Solution to Cholesterol Content and Size Distribution

**DOI:** 10.3390/pharmaceutics12111065

**Published:** 2020-11-09

**Authors:** Muhammad Ijaz Khan Khattak, Naveed Ahmed, Muhammad Farooq Umer, Amina Riaz, Nasir Mehmood Ahmad, Gul Majid Khan

**Affiliations:** 1Department of Pharmacy, Quaid-i-Azam University, Islamabad 45320, Pakistan; natanoli@qau.edu.pk (N.A.); farooqyarhussain@gmail.com (M.F.U.); aminaranjha@yahoo.com (A.R.); 2Department of Pharmacy, University of Swabi, Anbar, Swabi 23561, KPK, Pakistan; 3SCME, National University of Sciences and Technology (NUST), H12, Islamabad Postcode 4400, Pakistan; nasir.ahmad@scme.nust.edu.pk

**Keywords:** liposomes, simple methods, entrapped volume, PDI, Z-average diameter, Zeta potential, stability study, topography of liposomes, CI method, SPT method

## Abstract

Intricate formulation methods and/or the use of sophisticated equipment limit the prevalence of liposomal dosage-forms. Simple techniques are developed to assemble amphiphiles into globular lamellae while transiting from the immiscible organic to the aqueous phase. Various parameters are optimized by injecting chloroform solution of amphiphiles into the aqueous phase and subsequent removal of the organic phase. Further simplification is achieved by reorienting amphiphiles through a spontaneous phase transition in a swirling biphasic system during evaporation of the organic phase under vacuum. Although the chloroform injection yields smaller Z-average and poly-dispersity-index the spontaneous phase transition method overrides simplicity and productivity. The increasing solid/solvent ratios results in higher Z-average and broader poly-dispersity-index of liposomes under a given set of experimental conditions, and vice versa. Surface charge dependent large unilamellar vesicles with a narrow distribution have poly-dispersity-index < 0.4 in 10 μM saline. As small and monodisperse liposomes are prerequisites in targeted drug delivery strategies, hence the desired Z-average < 200 d.nm and poly-dispersity-index < 0.15 is obtained through the serial membrane-filtration method. Phosphatidylcholine/water 4 μmol/mL is achieved at a temperature of 10°C below the phase-transition temperature of phospholipids, ensuring suitability for thermolabile entities and high entrapment efficiency. Both methods furnish the de-novo rearrangement of amphiphiles into globular lamellae, aiding in the larger entrapped volume. The immiscible organic phase benefits from its faster and complete removal from the final product. High cholesterol content (55.6 mol%) imparts stability in primary hydration medium at 5 ± 3 °C for 6 months in light-protected type-1 glass vials. Collectively, the reported methods are novel, scalable and time-efficient, yielding high productivity in simple equipment.

## 1. Introduction

Liposomes are globular vesicles of amphiphiles with an aqueous core concealed in lipophilic lamellae. Phospholipids coalesce in a manner in which hydrophilic heads hold the water on both sides while hydrophobic-tails rootle away in the bilayers. The dual-compartment structure enables them to entrap both hydrophilic and lipophilic drugs in the core and the bilayer, respectively [1,2,3]. Liposomal classification is based on morphology (uni/multilamellar), dimensions (giant, large or small), surface-charge (anionic, cationic or neutral) and function (conventional, coated, targeted, triggered, stealth, slow-release or combinatorial) [4]. Liposomal dosage-forms enhance the stability and/or biodistribution of entrapped drugs [5,6]. However, the quality elements including size, morphology, surface charge, and ligand, if attached, essentially affect the pharmacokinetic behaviour of entrapped drugs [7,8]. Resemblance with bio-membranes [9], enhanced the permeability and retention effect [10], and/or active targeting [11,12,13] improved their pharmacological profile. Consequently, the liposomal drugs show enhanced therapeutic efficacy and safety [14,15] compared to conventional dosage forms. Therefore, liposomes are extensively employed in therapeutic and theranostic targeted drug delivery systems [3,16,17].

The structural resemblance with biological membranes upheld the liposomes as tempting biocompatible drug carriers [9]. Though paramount in multidrug targeting, the prevailing techniques hamper their primacy either due to sophisticated equipment, knotty techniques and/or residues of miscible organic solvent in the final product. Consequently, besides a variety of reported methods, scale-up to industrial manufacturing and scale down for personalized treatment remained challenging [18]. For this reason, the film hydration method (Bingham et al.; 1965) with or without modifications is still commonly used due to the simple procedure and equipment involved [2,5,16]. An organic solvent is either removed before hydration with aqueous phase (film hydration methods) or from the mixture of both phases (bulk methods) [4]. The rest is the manipulation of these basic principles for good results. Bulk methods include reverse-phase evaporation [19], organic-phase injection [20,21,22,23], solvent spherules formation [24], rapid preparation of giant liposomes [25], detergent dialysis [26,27], rapid solvent exchange [28], microfluidics [29,30], use of supercritical fluids [31] and modified electro formation [32]. The majority of these methods require sophisticated equipment and/or convoluted procedures that confine the commonness of the dosage-form.

Film hydration methods rout the film-forming components through intermediary solid phase [2,5]. Therefore, subsequent film hydration usually takes much longer and results in compositional inhomogeneity, particularly in the presence of cholesterol (Cho) [28]. Bulk methods are preferred for the preparation of liposomes due to ease, lamellar homogeneity and good entrapment efficiency. However, the organic phase plays a pivotal role in such methods for consigning phospholipids into the aqueous phase. Some of the bulk methods also used water-miscible organic solvents [21,25], which are difficult to be removed from the final formulation. The residues of such miscible solvents impede the stability of vesicles, denature the susceptible drugs and are also toxic to human health. Cho is an essential component to impart rigidity and hence physical stability to liposomes [33,34]. The absence of Cho in some of the bulk methods undoubtedly facilitated vesiculation but resulted in delicate vesicles and hence made them prone to leakiness in shelf-life storage.

Although they are indubitably preferred, the majority of the bulk methods are confined by using water miscible organic phase, convoluted procedures and/or need hi-tech equipment. Therefore, simple methods are reported here for instant preparation of liposomes with an immiscible organic solvent, high Cho contents, and the bypassing of intermediary solid film assuring homogenous lamellae [28]. Customarily, the methods are named as the chloroform-injection (CI) and spontaneous-phase-transition (SPT) methods. The CI method comprises the injection of amphiphiles and subsequent removal of immiscible organic-phase, while in the single-step SPT method, evaporation under reduced pressure from the biphasic mixture induced a de-novo reorientation of amphiphiles into lamellar vesicle. Complete removal of the organic phase is achieved due to immiscibility and lower boiling point through evaporation under reduced pressure. The CI method is advantageous over the other organic phase injection methods in terms of successful removal of immiscible organic solvent, lower poly-dispersity-index (PDI), smaller Z-average hydrodynamic diameter (Z-av), shorter processing time, use of lower temperature, high cholesterol contents and only simple equipment being required. Therefore, it is a method of choice for point of care formulations. However, injecting organic phase through a narrow bored needle limits its application in large scale production. On the other hand, the SPT method, besides having all of the above advantages of the CI method, overrides simplicity and handiness by excluding a cumbersome injection of organic phase and presenting whole material to vesiculation. The simultaneous addition of both immiscible phases and subsequent evaporation under reduced pressure imparts flexible scalability to the SPT method from the point of care to large scale production.

## 2. Materials and Methods

Hydrogenated phosphatidylcholine from egg yolk, EPC-3^®^, comprising Phosphatidylcholine, hydrogenated (98%), Phosphatidylethanolamine (0.1%), Lysophosphatidylcholine (0.5%), Sphingomyelin (0.1%) (HPCE) with a transition temperature of 55 °C, were gifted by Lipoid GMBH, Frigenstrass-4, Ludwigshafen, Germany. Cholesterol (Cho), Rhodamine-B (RhB), Chloroform (analytical grade) (CHCl_3_) were obtained from Sigma Aldrich. Nylon membrane filters (0.4 μ and 0.2 μ) (Sartorius, Germany), and dialysis membrane (12–14 kDa) (Membrane Filtration Products, Texas, USA) were used. Syringes with a needle of 0.16 d.mm (bore-dia) were purchased and their tips blunted with sandpaper. Freshly prepared double-distilled water (DW) was duly filtered with 0.1 µm membrane-filter prior to use. Stock solutions of HPCE/CHCl_3_ 1.00 mmol/mL and Cho/CHCL_3_ 1.00 mmol/mL were stored at 5 ± 3 °C.

### 2.1. Chloroform Injection Method

The required quantity of HPCE/Cho/CHCl_3_ solution was filled in the syringe with a 0.16 d.mm blunt needle. The aqueous phase was adjusted to the prerequisite temperature (25, 35, 45 or 55 °C) with high speed stirring by avoiding vortex formation to prevent the sedimentation and caking of solids at the base of the device. For a 12 mL aqueous volume 1.5, 3 or 6 mL of organic phase was promptly injected just below the surface of the aqueous phase in a 20 mL vial with a 2 cm diameter. The speed of the magnetic stirrer with a 1 cm bar was adjusted at 1000 rpm during the injection step. The mixture was allowed to mix at 10 rpm for next 2–3 min to obtain a uniform suspension. A milky suspension obtained was transferred to the rotary evaporator and allowed to evaporate under reduced pressure (through a 16-psi vacuum pump) at ∟45°, prerequisite temperature and 150 rpm for 20 min. A maximum of half of the rotating flask was occupied to allow sufficient surface area for evaporation.

During optimization studies, unreacted material and/or giant vesicles were removed through slow centrifugation at 25 °C, 1000 *g* for 3 min with zero acceleration/deacceleration. However, optimized formulations were sized through serial membrane filtration immediately while maintaining the process temperature (Section 2.3). Replicate trials were produced to ensure the reproducibility and quality of vesicles. Various formulation and process parameters were optimized including variable quantities of the organic phase, aqueous phase, Cho concentration and temperature while the quantity of phospholipids was kept constant.

### 2.2. Liposome Preparation by SPT Method

Chronologically, in order to improve efficiency in terms of processing time, yield and/or material loss, the optimized parameters of the CI method were further investigated through the SPT method. DW was brought to the required temperature in the rotating flask of a rotary evaporator. The solution of HPCE/Cho/CHCl_3_ was gently added, partitioning at the bottom of the flask. The mixture was allowed to rotate at ∟45°, 150 rpm for 20 min under reduced pressure at 45 °C to evaporate the immiscible organic phase. Liposomes were produced de-novo by spontaneous reorientation of amphiphiles while transiting from the organic to the aqueous phase. The vesicles produced by the SPT method were evaluated and compared with the CI method under a given set of conditions. Highly concentrated large unilamellar vesicles (LUVs), comprising HPCE/DW 4 µmol/mL were sized through serial membrane filtration at the processing temperature (Section 2.3).

### 2.3. Sizing and Purification

Sizing of liposomes was obligatory to obtain the desired size and narrow distribution by serial membrane filtration [35,36]. Briefly, the freshly prepared liposomes were passed 5x each through 0.4 µm followed by a 0.2 µm nylon membrane filter. Sizing was performed in the primary hydration medium while maintaining the processing temperature.

The sized vesicles were purified through membrane dialysis to remove non-liposomal contents if required. Concisely, the sized vesicles in primary hydration volume were transferred to the dialysis sac (12–14 kDa MW cut-off) and placed in 10x freshly prepared distilled water. The temperature was maintained at 5 ± 3 °C and stirring was conducted at 10 rpm for 24 h. Washing media was replaced every 6h to ensure equilibration. The product was collected and stored in type-1 glass vials duly wrapped in aluminium foil at 5 ± 3 °C until further use.

### 2.4. Dynamic light scattering

The formulations were analysed for Z-av, PDI and zeta potential (ζ-potential) using the dynamic light scattering (DLS) technique through Zeta-sizer ZS90 (Malvern Instruments, Malvern, UK). Size analysis was carried out at 25 °C, scattering-angle 90°, with He-Ne laser at λ 632.8 nm in aqueous dispersion at 25 °C with 1.33 RI, 1 cps viscosity and a dielectric constant of 79. For comparative size distribution and to avoid multiscattering, 100× dilution in DW at 25 °C was used. Subsequently, to minimize the effect of surface potential, 100× dilution of liposomal stock dispersion, in 10 µM NaCl (clarified through 0.2 µ membrane filter), was also analysed for comparative size distribution.

### 2.5. Scanning Electron Microscopy

A scanning electron microscope (SEM) (JSM 6490A, JEOL, Tokyo, Japan) was used to study and compare the size and morphology of liposomes. Unsized samples produced by the CI and SPT methods under the same set of optimized parameters (Section 2.1 and Section 2.2) were subjected to comparative analysis. A quantity of 50 µL from 100× diluted stock dispersion was dropped on the silicon wafers and gently blotted after 2–3 min. Surface water was evaporated at room temperature for 5 min. The dried smear was gold-coated in Joel 1100 (JFC Ion Sputter) at 50 mA in argon environment (50 Pa) for 50 s. Coated samples were observed at 10,000× magnification under the SEM mode.

### 2.6. Atomic Force Microscopy

The topography of optimized formulations after sizing was performed with a Scanning Probe Microscope (SPM) (JSPM-5200, JEOL, Japan). Imaging was performed at 20–25 °C in DW on 1 cm^2^ silicon wafers. A quantity of 50 µL of 100× diluted stock dispersion was dropped on a silicon wafer to minimize clustering and sticking of the vesicles. The samples were gently blotted after 2 min with filter paper. Excessive water from the smear was evaporated at room temperature before loading samples on the specimen stage. All samples were observed within 5–15 min of deposition. The atomic force microscopy (AFM) was operated in amplitude-detection (AC-AFM) mode in atmospheric conditions. Aluminium back-coated silicon nitride tip was used for scanning. AC-AFM scans the sample by vibrating the cantilever with constant amplitude in an area of a specific frequency at a distance of 5–20 Å [37]. The change in the voltage (error signal) for a constant amplitude of cantilever is computed into an image. The cantilever was moved to approach the sample to a point where inter-atomic forces interact to scan the sample, and confirmed through the reference voltage and Z-piezo position. Images were scanned and analysed in scanning and processing mode, respectively, through WINSPM 5 software. Topographical images along with error signals were obtained, showing minor superficial variation in the samples. The diameter and height of the vesicles were measured by drawing a line with pointers across the images. Surface roughness was directly estimated from the arithmetic average roughness (Ra) values in height mode. The volume of vesicles was calculated by using the following formula:*V* = 1/3 *μh* (3*x*^2^ − *h*^2^)(1)
where “*r*” represents the radius and “*h*” is the height of the vesicles.

The rigidity of liposomes is the measure of resistance to deformation in a set of given circumstances. In-vitro rigidity of liposomes was calculated from the height data of AC-AFM and size data from DLS using 10 µM NaCl as a dilution medium to compress the surface charge cloud [38]. Average rigidity of the samples was calculated through the following equation:Rigidity of liposomes = H/d(2)
where “*H*” is the height from AFM data and “*d*” is the diameter of vesicles from DLS data.

### 2.7. Entrapped Volume

Aqueous volume entrapped by liposomes is indicative of the process efficiency and consequent encapsulation of dissolved drugs. RhB was dissolved in DW at a concentration of 500 µmol/mL. This solution was used to prepare triplicate samples of liposomes at variable concentrations of HPCE/CHCl_3_ in a constant volume of the aqueous phase using optimised conditions (Section 3.2). The RhB-liposomes were sized through serial membrane filtration and purified by exhaustive dialysis (Section 2.3). A quantity of 100 L of dialyzed liposomes was added to 3 mL of DW containing 2% Triton X-100 (lysing solution) to avoid liposomal turbidity and relieve the self-quenched RhB. The mixture was transferred to a quartz cuvette and absorbance was measured at λ-max 550 nm while taking DW as a blank. RhB solution without phospholipids was analysed to calculate any traces of unentrapped RhB. The observed values were quantitated against a standard curve of RhB solution in lysing solution. Each sample was also analysed for phospholipid concentration by the Bartlett method using colorimetric determination of inorganic phosphate. However, no significant loss was observed after multiple analysis. Entrapped volume was calculated by:μL. mol^−1^ = (*V* × *Am*)/*MAs*(3)
where “*M*” represents a number of moles of phospholipids in 3 mL dialyzed liposomes, “*Am*” is the measured absorbance of RhB in emulsified sample and “*As*” denotes standard absorbance of RhB in known volume *V* (μL) [23].

### 2.8. Accelerated Stability Studies

International Conference on Harmonization (ICHQ) protocol Q1A(R2)2.2.7.4 was followed to test the accelerated stability of liposomes intended to be stored in refrigeration [39]. The liposomal preparation containing HPCE/Cho as 4:5 molar ratios and HPCE/DW 4 µmol/mL (Section 2.2 and Section 2.3) were exposed to accelerated storage conditions. The liquid samples in primary hydration media were stored in Type-1 glass vials aptly wrapped in aluminium foil and each kept at room temperature, 23 ± 2 °C, and in a refrigerator, 5 ± 3 °C, for 6 months. Being stored in impermeable and light-resistant containers, only thermal stability was a concern in terms of probable instability. Additional test points were included to observe any change in a small interval of time including day 0, 30, 60, 90 and 180. Physical appearance in terms of uniformity or precipitation/aggregation was considered decisive to the stability of various samples. Samples at a given interval were analysed for Z-av and PDI (Section 2.4).

### 2.9. Data Reporting

All experiments were performed in replicate with a variable n value. Optimization experiments were reproduced in duplicate while optimized formulations were reproduced multiple times, to ensure reproducibility, and as required to study various parameters. Data were averaged and presented as ± standard deviation.

## 3. Results

### 3.1. Key Outcomes of Optimization Studies

The samples were assessed based on the Z-av, PDI and physical uniformity during optimization studies. In the same set of experiments, kilo count per second/count-rate (kcps) during DLS measurement was considered as a measure of comparative vesiculation. Smaller size, PDI and higher kcps at given sample dilution were considered as parameters of attainment.

#### 3.1.1. Optimization of the Organic Phase

Liposomes were prepared through the CI method (Section 2.1) at 25 °C and HPCE/DW 0.15 µmol/mL. Z-av slightly declined within the same set of experiments by diluting phospholipids in the organic-phase (Table 1a) HPCE/DW 8–2 µmol/mL. Further dilution yet again resulted in a slight increase in size. However, both PDI and kcps continuously improved with increasing dilutions of HPCE/CHCl_3_. Another set of experiments (Table 1d) was performed using HPCE/DW 0.38 µmol/mL and variable HPCE/CHCl_3_ to evaluate the minimum required volume of organic-phase. All the above-mentioned parameters increased with increasing concentrations of HPCE/CHCl_3_ from 10 to 30 µmol/mL, while precipitation was observed at HPCE/CHCl_3_ 60 µmol/mL.

#### 3.1.2. Optimization of the Aqueous Phase

The CI method at 25 °C, HPCE/CHCl_3_ 8 µmol/mL and variable aqueous volume were used. The vesicle size was reduced with increasing concentrations of solids in the aqueous phase (Table 1b) until HPCE/DW was 0.38 µmol/mL, while kcps were markedly increasing towards concentrating aqueous suspensions. The CI method at 25 °C was effective in diluting HPCE/DW ≤ 0.38 µmol/mL. Further increasing of solids resulted in precipitation and the sticking of material to the walls of the rotating flask.

#### 3.1.3. Optimization of HPCE/Cho Molar Ratio

The liposomes were prepared by the CI method at 25 °C, HPCE/DW 0.38 µmol/mL and HPCE/CHCl_3_ 8 µmol/mL. The HPCE/Cho ~4:5 molar ratio exhibited better results in terms of uniformity, smaller size, and PDI (Table 1c) without treatment for sizing. The lower molar concentration of Cho showed broader distribution and hydrodynamic diameter while a higher ratio exhibited precipitation against the pre-set quantity of HPCE.

#### 3.1.4. Optimization of Processing Temperature

Liposomes were formulated at variable temperatures of 25, 35, 45 and 55 °C by the CI method with the HPCE/Cho 4:5 molar ratios and a high solids/solvent ratio (Table 1e). Preoptimized concentrations of HPCE/DW 4 µmol/mL, HPCE/CHCl_3_ 7.5, 15 or 30 µmol/mL were evaluated at the above-mentioned temperatures. Precipitation was observed at 25 °C and 35 °C, while at 55 °C aggregation was observed with diluted HPCE/CHCl_3_ 7.5 µmol/mL. Uniform suspension of vesicles with higher kcps was obtained at 45 °C, i.e., below the transition temperature of lipid employed. Serial dilutions of HPCE/CHCl_3_ were directly related to the smaller size vesicles at any given temperature and volume of the aqueous phase.

### 3.2. Parameters of Optimised Formulation

The optimized formulations comprised HPCE/Cho 4:5 molar ratios, HPCE/DW 4 µmol/mL, HPCE/CHCl_3_ 7.5, 15 or 30 µmol/mL (variable concentration) or DW/CHCl_3_ 16:1, 4:1 and 2:1 *v*/*v*. Cho/DW 5 µmol/mL was used as the final concentration by dissolving together with HPCE in the organic phase. Optimized conditions included 45 °C temperature followed by rotary evaporation under reduced pressure (using 16 psi vacuum pump) at ∟45° and 150 rpm. Different protocols for the SPT and CI methods have already been explained in Section 2.1 and Section 2.2, respectively. The liposomes produced by both methods were sized and purified (when required), as mentioned in Section 2.3. Optimized parameters were decided based on lower PDI, Z-av, physical uniformity and high solid to solvent ratio at a possible lower temperature, as mentioned in Section 3.1.1, Section 3.1.2, Section 3.1.3 and Section 3.1.4. Various problems encountered during experimentation were effectively resolved and are summarized in Table 2.

### 3.3. Comparison of CI and SPT Methods

For comparative studies, vesicles were produced through both the CI and SPT methods (Section 2.1, Section 2.2). Optimized parameters were used with 2× increasing concentration of HPCE/CHCl_3_ ≥ 7.5 µmol/mL in organic-phase against a pre-set quantity of HPCE/DW 4 µmol/mL (Section 3.2). The SPT method was a further simplified form of the CI method, bypassing the cumbersome step of organic phase injection (Figure 1a,b) and subjecting the whole material for vesiculation.

#### 3.3.1. Size Distribution Analysis

Both the CI and SPT methods yielded comparable results in terms of the Z-av and PDI of the vesicles. However, slightly higher Z-av and PDI values were observed through the SPT method compared to the CI method (Figure 1c–f) under the same set of optimized parameters. The formulations produced with the SPT method were more turbid and gave higher kcps over 100× dilution than the CI method. A low solids/solvent ratio in the organic-phase resulted in lower Z-av and PDI values of the vesicles against a pre-set HPCE/DW of 4 µmol/mL. The results showed a parabolic pattern with the smallest size at DW/CHCl_3_ 4:1 or HPCE/CHCl_3_ 15 µmol/mL as compared to 30 µmol/mL and 7.5 µmol/mL, while slight variation in PDI was observed within the same group of experiments.

#### 3.3.2. Effect of Sizing by Serial Filtration

Size reduction by membrane filtration (Section 2.3) resulted in lower PDI and Z-av values (Table 3). The size of filtered vesicles was consistent with the pore size of the final membrane used for filtration. The lower kcps for the same set of experiments with 100× dilution was indicative of a loss in the number of vesicles per mL of the aqueous phase. Since sizing down by membrane filtration was necessary to obtain the desired size and PDI, the SPT method was preferred for better yield, processing time, convenience and hence cost-effectiveness.

#### 3.3.3. Scanning Electron Microscopy

The morphological analysis under SEM revealed a circular appearance with smooth surfaces of liposomes produced by both the CI and SPT methods (Figure 1d,e). A thick walled outer boundary without any concentric layers confirmed a unilamellar structure covered by the surface membrane. However, SEM images showed an obvious difference in size amongst CI and SPT liposomes without sizing. SPT liposomes were larger in size and had a comparatively broader distribution of 1023.18 ± 336.44 d.nm, while CI liposomes showed 581.71 ± 61.22 d.nm, being comparatively smaller in size and more uniform than SPT liposomes.

### 3.4. Properties of Optimized Liposomes

Both the CI (Section 2.1) and SPT (Section 2.2) methods under optimized conditions (Section 3.2) yielded LUVs. The LUVs were sized by serial membrane filtration (Section 2.3) for narrow size distribution. Sized liposomes were consistent with the smallest pore size of the membrane used for final filtration, and lower PDI and size-dependent variable ζ-potential (Table 3). After careful handling, a 16.7% loss was observed to the initial volume of the aqueous phase used. Half of the loss was attributed to the evaporation of water under reduced pressure signifying complete removal of the organic phase.

#### 3.4.1. Size Distribution and Surface Charge Analysis

The SPT method, under optimized conditions (Section 3.2), yielded LUVs with Z-av ≤ 1466±28.84 d.nm and PDI ≤ 0.39 ± 0.003, dependent on varying concentrations of HPCE/CHCl_3_. The CI method also furnished LUVs, but of comparatively smaller size ≤ 627.4 ± 0.99 d.nm and PDI ≤ 0.37 ± 0.020 (Figure 1c,d). More handling losses were observed due to the injection step of the CI method (Figure 1a,b). DLS analysis in DW and 10 µM NaCl showed different but comparable results indicating ζ-potential dependent size variation (Table 3). The Z-av of unsized liposomes declined with further dilution of solids in an organic solvent for a pre-set quantity of the aqueous phase (Table 3). PDI and ζ-potential of unsized vesicles showed a slight variation in parabolic pattern with a minimum at the centre of the variable concentrations in the organic-phase. A maximum PDI of 0.579 and a minimum ζ-potential of −20.4 mV were observed at the highest dilution of HPCE/CHCl_3_ 7.5 µM/mL using distilled water as dilution medium. The variable Z-av and ζ-potential values at variable HPCE/CHCl_3_ concentration showed a direct relationship (Table 3). The PDI of vesicles after sizing was significantly improved in the range of 0.15 to 0.095. The ζ-potential of vesicles was slightly increased but more uniform after sizing in the range of −18.15 mV to −17.1 mV based on size-dependent variation.

#### 3.4.2. Atomic Force Microscope Topography

Liposomal stock dispersion after 100× dilution in DW was imaged through atomic force microscopy in amplitude-detection mode (AC-AFM). The stock dispersion contained HPCE/DW 4 µmol/mL and Cho 55.6 mol% of total solids. Liposomes were observed within 5–10 min of deposition on silicon-wafers. The promptness in observation was critical to avoid topographical variations due to substrate–sample interaction and/or environmental effects [40]. Spheroidal vesicles having size-dependent surface-homogeneity signified aptness of the sample preparation method and the scanning technique used (Figure 2). However, a probable stretching effect of the cantilever, sample–substrate interaction and flaccidity of the liposomes lent a difference in observed length to width ratio (Figure 2b,e) [41]. The vesicles maintained their integrity due to lamellar elasticity allowing a slight bend during the scanning process. The observed diameter of the sized vesicles was in accordance with the pore size of the smallest filter-membrane (Table 3). The height of the samples was accorded to the size and rigidity of the samples. A slight difference in the height of the sized samples compared to corresponding unsized vesicles was observed. Mean diameter, height, average surface roughness (Ra) and rigidity were improved by reducing the size of any given sample (Figure 2, Table 3). Surface homogeneity was confirmed from 3D topographic images analysed in the process mode of an inbuilt software (WINSPM 5). The section view and three-dimensional imaging plot indicated a smooth and regular surface profile of the observed liposomes. The surfaces of sized vesicles appeared more uniform than the corresponding unsized samples (Figure 2c,f). In vitro rigidity of liposomes was assessed from the height data of individual vesicles through AC-AFM in combination with Z-av measured after 100x dilution in 10 µM NaCl. The data of Z-av in saline medium resembled the length of vesicles measured through AFM, hence, they were considered suitable to assess rigidity. Comparing various concentrations of HPCE/CHCl_3,_ 15 µmol/mL sized samples showed maximum rigidity of 0.14 ± 0.03, height of 30.58 ± 6.82 nm, Z-av of 149.25 ± 30.2 nm, and roughness of 5.15 ± 1.68 nm. The calculated volume was size-dependent and progressively increased with the decreasing molar concentration of amphiphiles in the organic-phase. However, more uniformity in volumes was observed at HPCE/CHCl_3_ 15 µmol/mL of the unsized vesicles. The majority of the liposomes appeared intact and spheroidal in the obtained images. However, slightly flattened structures were observed due to the flaccid nature of liposomes obvious from their length to height ratio (Table 3 and Figure 2).

#### 3.4.3. Entrapped Aqueous Volume.

Large entrapped aqueous volumes were observed to be consistent with the calculated volumes (Table 3). A slight variation in variable HPCE/CHCl_3_, ranging from 20.09 ± 0.18 to 23.87 ± 0.18 L/mol, also reflected size dependence. Unsized liposomes were excluded from the study due to broader PDI, and hence were of little practical implication. However, the long dialysis period of 24 h may have some leakage effects which were not included in the calculations. So, the given values might indicate a minimum achievable entrapped volume of the available aqueous phase.

#### 3.4.4. Accelerated Stability Studies

The liposomal suspension remained uniform in type-1 glass vials throughout the study. However, visible precipitates/aggregates appeared in the plastic falcon tubes for the same set of experiments and hence were excluded from the study. During the first month of storage at both temperature conditions in type-1 glass vials, a slight increase both in PDI and Z-av was observed (Figure 3d). However, the values of PDI gradually declined but Z-av increased over an extended period of storage, showing aggregation of vesicles. Samples stored in the refrigerator showed a minute increase in size from 226.3 nm to 246.6 nm and PDI from 0.105 to 0.185 (Figure 3a,b) over a period of 6 months. However, a comparatively more significant change was observed at room temperature where Z-av increased from 226.3 nm to 327.5 nm and PDI from 0.105 to 0.214 (Figure 3a–c).

#### 3.4.5. Process Validation

The batch sizes of optimized formulations were scaled-up by 5× increment and evaluated for reproducible results. A variety of phospholipids based on their hydrogenation and source material were used to prepare liposomes under an optimized set of parameters and checked for reproducible results (data not included in this article). All formulations were produced by using optimized formulation and processing parameters (Section 3.2). The three reproducible trial formulations and 5x scaled-up batches yielded comparable results in terms of Z-av and PDI.

## 4. Discussion

### 4.1. Key Features of Optimization Studies

The experimentation aimed to use a higher solids/solvent ratio at possibly lower temperature, imperative to high encapsulation efficiency, particularly of thermosensitive molecules [5]. Samples produced at 45 °C showed uniform dispersion while using a high solid/solvent ratio. Z-av of optimized formulations gradually decreased with the declining molar concentration of HPCE/CHCl_3_ µmol/mL and vice versa [24,42] against fixed aqueous volume. It is presumed that the slow movement of diluted phospholipids from organic to aqueous phase favoured the formation of smaller sized vesicles, whilst the PDI was slightly parabolic from high to lower HPCE/CHCl_3_ µmol/mL, suggesting that DW/CHCl_3_ at 4:1 or HPCE/CHCl_3_ at 15 µmol/mL was a suitable concentration for comparable smaller size and PDI of liposomes. This suggested that lower temperatures and concentrated amphiphiles in the organic-phase collectively resulted in condensation and tactless relocation into the aqueous phase, while, at higher temperatures (55 °C), although faster removal of organic phase was favoured, the solid components were tendered in the straightened molecular dimension or GUVs and aggregation consequently occurred. The concentration of Cho was found to be critical for vesicle formation with an optimal level of HEPC/Cho as a 4:5 molar ratios The concentration of Cho below this level promoted vesicle aggregation if not sized, while the above levels favoured precipitation [43,44]. The sizing of liposomes by serial membrane filtration is associated with concomitant deformation and reformation of vesicles during extrusion through narrow orifices of the filter membrane. Therefore, the extrusion of LUVs in the primary hydration medium is essential for the restoration of dissolved components [36].

### 4.2. Properties of Optimized CI and SPT Liposomes

Evading the injection phase of the CI method resulted in the SPT method without any significant variation in the quality of liposomes, but furnished process efficiency (Figure 1a). Therefore, the SPT method may be preferred for its further simplicity, handiness, least interaction of organic solvent to aqueous components and better yield. A high solid solvent ratio signified high encapsulation efficiency of aqueous drugs [5], while immiscible organic solvent gave true molecular dispersion of organic components, ensuring good entrapment of lipophiles in the lamellar compartment. Attempts were made to analyse the residual traces of CHCl_3_ in the final product but could not obtain reproducible results after 10 min of rotary evaporation under vacuum. However, its complete removal is evidenced from the previous reports [25,28], as is the overall volume loss of total aqueous suspension (Section 3.4). The desired size distribution required essential steps of sizing [35]. The CI liposomes can be produced over a wide range of temperatures (25–45 °C) but require larger aqueous volumes at low temperatures signifying low encapsulation efficiency of hydrophiles. The liposomal size gradually increased with increasing phospholipids concentration in both or any solvent phase [6,30]. The aggregation of vesicles at 55 °C (hydrated transition temperature of HPCE) infers the formation of larger globules due to the high fluidity of phospholipid molecules. Cho 55.6 mol% yielded uniform vesicles while lower concentrations indicated phase segregation [43,44]. High Cho contents not only aided the shelf-life stability of liposomes but efficient encapsulation of aqueous drugs as well [33,34].

Size and surface charge on liposomes are imperative in the stability, kinetics, biodistribution, enhanced permeation and retention, sterilization and interaction with the targeted cells [45]. Usually, the liposomes are sized by membrane filtration at above the phase transition temperature of lipids [35]. The pore size of the membrane used for extrusion determines the size of vesicles with more uniformity compared to other methods [8,36]. In this study, the liposomal size was effectively reduced at the processing temperature that was ≤10 °C below the phase transition temperature of the lipid in use. By observing the above mentioned formulations in 10 µM NaCl by DLS and topographical images of AC-AFM (Table 3, Figure 2), the ζ-potential dependant size distribution in DW is indicated. The type of hydration medium for the production of vesicles plays a crucial role in size distribution where DW lent smaller size and broader PDI [46]. The aim of using DW as a hydration medium in this study was to investigate various parameters in the very raw form to leave room for improvement while working with drug-loaded formulations.

AFM is the most suitable non-invasive technique to observe the liposomes and soft biological samples without staining, under atmospheric conditions [37,47], based on its simplicity and high resolution approaching to 1 Å [40], while both contact and non-contact AFM have the drawbacks of damaging soft vesicles and low resolution, respectively [37]. AC-AFM exerts low pressure while operating at the separation of 5–20 Å and high resolution by intermittent tapping. Variation in the height to width ratio of AC-AFM images depicted the size-dependent rigidity and flaccid nature of the vesicles [38]. The anionic vesicles deposited on the silicon wafers with a slight negative surface-charge reduced the sample–substrate interaction [48]. The diameter measured by AC-AFM was smaller than the hydrodynamic diameter through DLS and was in accordance with the smallest filter-membrane used for sizing. The larger Z-av diameter was electrical potential dependent with ζ-potential −17.3 ± 0.6 mV. Other electron microscopy techniques require staining of samples and observations under high vacuum which affect the delicate structure of liposomes. The coating process for SEM topography collapsed the flaccid liposomes from globular to circular objects. The unilamellar images with sharp boundaries were slightly larger in diameter than the AFM images. This method gives gross estimation of lamellarity by staining interlamellar spaces and resolving the lamellar walls [49,50]. Rigidity is an important factor aiding the stability of liposomes, particularly if a long systemic circulation time is required. High Cho contents, the nature of phospholipids, surface potential and coating can aid the rigidity of a given size of liposomes [38].

High encapsulated concentration is beneficial to reduce the dose size, increased dosing interval and make the product cost-effective. Ideally, a 130 nm liposomes entrapped volume will be 36 L/mol; by ether injection, it was reported 14 ± 6 L/mol. [20] HPCE/Cho 50 mol% gave 16.41 L/mol and filtered 0.22 µm filter gave 9.0 L/mol. [23,24] In current methods, the de-novo assemblage and high solid/solvent ratio in aqueous phase favoured large entrapped volume (Table 3), ensuring high encapsulation of hydrophiles [51].

### 4.3. Vesiculation Pathway

The HPCE/Cho mixture yielded a true solution in CHCl_3_. In the CI method, the amphiphiles spontaneously reoriented in the aqueous-phase, leaving the orifice of the needle. The polar head interacted with water molecules while nonpolar tails sequestered in the bilayer formed by the adjacent chains of amphiphiles [52]. The resultant highly unstable milky suspension was a mixture of vesicles and micelles containing both solvents. Removal of immiscible organic solvent stabilized the lamellar globules by holding water on both sides. However, for concentrated solid/solvent dispersions, a minimum temperature of 45 °C helped to prevent precipitation before assemblage into the lamellar vesicles.

In the SPT method, organic-phase containing amphiphiles was partitioned at the bottom of the flask. CHCl_3_ readily exchanged solute molecules while moving through the aqueous-phase during rotary evaporation. Amphiphiles reoriented on contact with water to minimize surface free energy by sequestering hydrophobic tails in the bilayers [52]. The size of the vesicles was a function of fragmentation achieved through swirling of the aqueous phase and the rate of evaporation of organic solvent to exchange phospholipids. This phenomenon was evidenced by larger size vesicles due to gradient solid/solvent ratio in organic-phase (Table 3).

### 4.4. Stability of the Liposomes

Both the CI and SPT methods spontaneously relocate HPCE/Cho from organic to aqueous environment, indicating uniform lamellar distribution [28], aiding the stability of liposomes. Cho enhances the strength of liposomes by modifying fluidity and interaction of phospholipids in the lamellar membranes [14,40]. The saturated acyl chain of HPCE has strong interaction with the steroidal ring of the Cho molecule. Hence, Cho concentration (55.6 mol%) showed high compressibility and low permeability to water [1,34] during storage. Therefore, the sole reason for stability might be a high molar concentration of Cho [38]. Storage at 5 ± 3 °C restricted molecular movement and hence stronger lamellae [15]. A size-dependent ζ-potential −17.1 ± 0.6 mV also aided in the stability of formulation by exhibiting inter-vesicular repulsion [6]. A slight increase in the size of the liposome at both temperatures indicated swelling of the bilayers while stored in the primary hydration medium. Overwhelmingly, the shelf-life stability can be improved by further enhancing the ζ-potential and lyophilization with suitable cryoprotectant [5,6].

### 4.5. Scalability

The injection phase of the CI method may restrict the method for small batches until some mechanical device for injection is adapted. However, the SPT method was more flexible to be scaled up and down due to the simple manufacturing protocol and equipment required for production. Scale-up studies (5×) showed reproducible results from 12 mL to 300 mL batch sizes. A maximum of half of the rotating flask of the rotary evaporator may be occupied to allow sufficient surface area for evaporation.

### 4.6. Pros and Cons of the Methods

The methods described in this paper were advantageous in terms of swiftness, reproducibility, productivity, adaptability, homogeneity and larger entrapped volume. The process loss of the CI method was effectively compensated in the SPT method, where the whole material was subjected to vesiculation. However, a comparatively smaller size and PDI were the overriding features of the CI over the SPT method. The use of immiscible organic solvent and vacuum evaporation facilitated easy removal of organic-phase and minimised interaction with aqueous components, benefiting sensitive molecules. Use of lower temperature, i.e., ≤10 °C below the phase transition temperature of lipids, and short processing time will aid in the encapsulation of thermolabile drugs. Processing at 25 °C was possible in large aqueous volume but the loss of diluted solutes and hence low encapsulation efficiency is obvious. The liposomes may be used fresh or stored in light protected type-1 glass vials in the liquid form for a longer period at 5 ± 3 °C. However, the potential disadvantages include cumbersome size reduction by serial membrane filtration to obtain smaller size and PDI. The unentrapped material if not desired must be removed by washing through dialysis at 5 ± 3 °C to avoid aggregation and consequent size growth of the vesicles.

### 4.7. Comparison with Other Methods

The current development aimed at creating simple method in terms of required equipment, processing time, versatility and handiness while preparing high-quality liposomes. Unlike film hydration methods [2,5,13], intermediate solid phase formation was bypassed, ensuring compositionally homogeneous and stable lamellae besides using high molar concentrations of Cho (55.6 mol%). A larger entrapped volume than those of film hydration methods was ascribed to the spontaneous rearrangement of phospholipids. Swiftly sizing by membrane filtration of CI and SPT liposomes within the same hydration medium was advantageous for other size-reduction methods like sonication and homogenization [36] in retaining the entrapped entities.

Bulk methods give comparable results to film hydration methods in terms of quality but have several limitations. Large aqueous volumes were used in the majority of methods, leading to high entrapped volume but low encapsulation efficiency [20,21]. Contrariwise, the CI and SPT methods used concentrated solutions and bulk methods to overcome the aforementioned limitations. Similarly, the use of miscible organic solvents and/or high temperature [20,21,22,25,27] limited some methods to stable drugs under such conditions, with a risk of leakiness. Exclusion of Cho [21,25] certainly favoured vesiculation at lower temperatures but led a compromised physical stability and lower entrapment efficiency [34]. Alternatively, the CI and SPT methods employed high Cho contents in immiscible organic solvent and temperature ≤ 10 °C below the transition temperature of the lipid used. The CI method was applicable at lower temperatures (25 °C) but higher aqueous volumes limited its suitability due to low encapsulation of diluted hydrophiles. Even though some methods employed immiscible organic solvents yet high lipid concentrations and highly sophisticated equipment and/or procedures confined their commonness [24,28,29,30,31,32]. Instead, the CI method needed an injection system followed by rotary evaporation but, even so, the SPT method eliminated the injection step from the process.

## 5. Conclusions

Collectively, both the CI and SPT methods helped to attain broader solutions for a variety of parameters for the preparation of liposomes, such as composition, temperature, Z-av, and scalability. Simple techniques and equipment were used to obtain the desired size distribution. A processing temperature 10 °C below the phase transition temperature of different phospholipids with 55.6 mol% of Cho was effective in formulating high solid/solvent ratios like HPCE 4 μmol/mL of aqueous phase. Processing temperature was lowered further by using diluted solutions of solids in any or both phases, facilitating the encapsulation of thermophiles. However, the minimum possible volume of immiscible organic phase was always preferred to ensure effective removal to safeguard the shelf life of liposomes and quality of entrapped entities. High solid/solvent ratios resulted in larger size vesicles but were imperative to higher entrapment efficiency of hydrophiles. Therefore, sizing at processing temperature in the primary hydration medium was pivotal to retain the initial concentration of entrapped molecules while attaining the desired Z-av and PDI. Lamellar homogeneity together with high cholesterol contents and absence of residual immiscible organic solvent aided the shelf life stability. The produced liposomes were stable as per ICH guidelines at 5 ± 3 °C while packed in light protected type-1 glass vials. However, lyophilization with suitable cryoprotectant, depending upon the nature of the phospholipids and entrapped molecules, will definitely enhance the shelf life quality of the product. The use of immiscible organic-phase assisted in entrapping both lipophilic and hydrophilic molecules simultaneously, signifying its equal appliance in diversified fields using nano formulations. Validation and scale-up studies ensured the handiness of the SPT method for a variety of amphiphiles, and scaleup for commercial production or down to point of care use. However, the CI method is currently restricted to small batches only until suitable mechanical equipment is adapted. Entrapment of various lipophilic and hydrophilic molecules and their stability in the intended dosage-forms need to be evaluated. Once the simple equipment is set up, the LUVs can be obtained within 20 min followed by sizing if needed. This study therefore demonstrates a scalable, simple and hence cost-effective technique in the formulation of liposomes.

## Data Availability Statement

All data generated and analysed during the current study are included in this published article. Further details and support are available from the corresponding author on reasonable request.

## Figures and Tables

**Figure 1 pharmaceutics-12-01065-f001:**
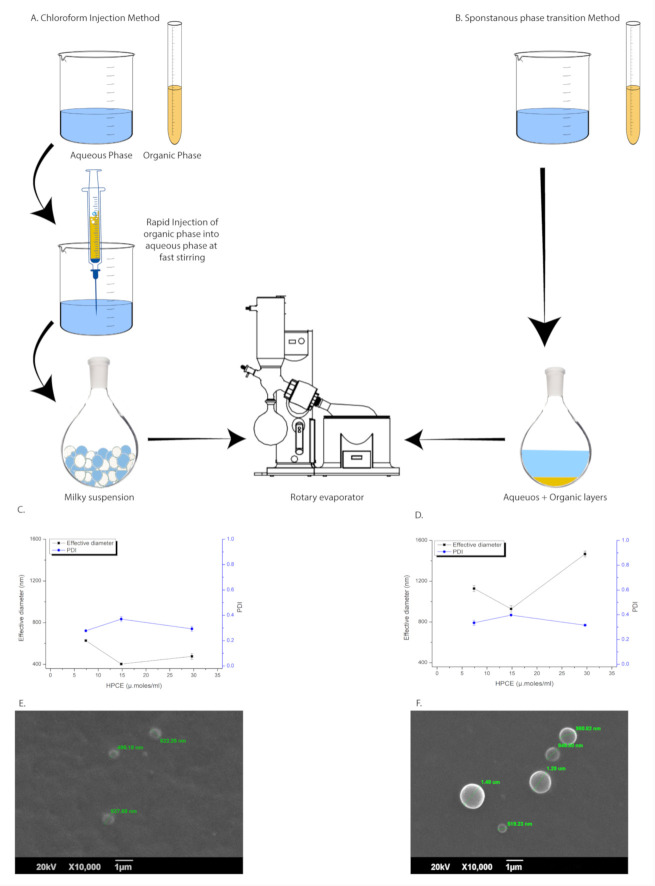
Comparison of CI and SPT liposomes produced with optimized parameters and without sizing. (**a**): CI method, (**b**): SPT method, (**c**): Z-av and PDI of unsized CI-liposomes, (**d**): Z-av, and PDI of unsized SPT-liposomes, (**e**): SEM image of unsized CI-liposomes, (**f**): SEM image of unsized SPT-liposomes.

**Figure 2 pharmaceutics-12-01065-f002:**
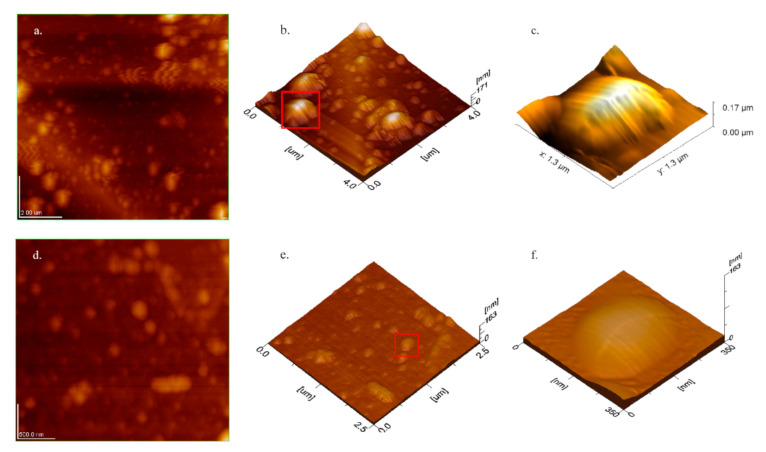
AFM topography of liposomes produced with optimized parameters showing comparison between sized and unsized liposomes. (**a**): 2D image of unsized liposomes, (**b**): 3D image of the same sample, (**c**): 3D image of selected single liposomes showing larger size, more height and surface roughness, (**d**): 2D image of sized liposomes, (**e**): 3D image of sized liposomes, (**f**): 3D image of selected single liposomes showing flattened and smaller size and size-dependent lower height but smooth surface and high rigidity.

**Figure 3 pharmaceutics-12-01065-f003:**
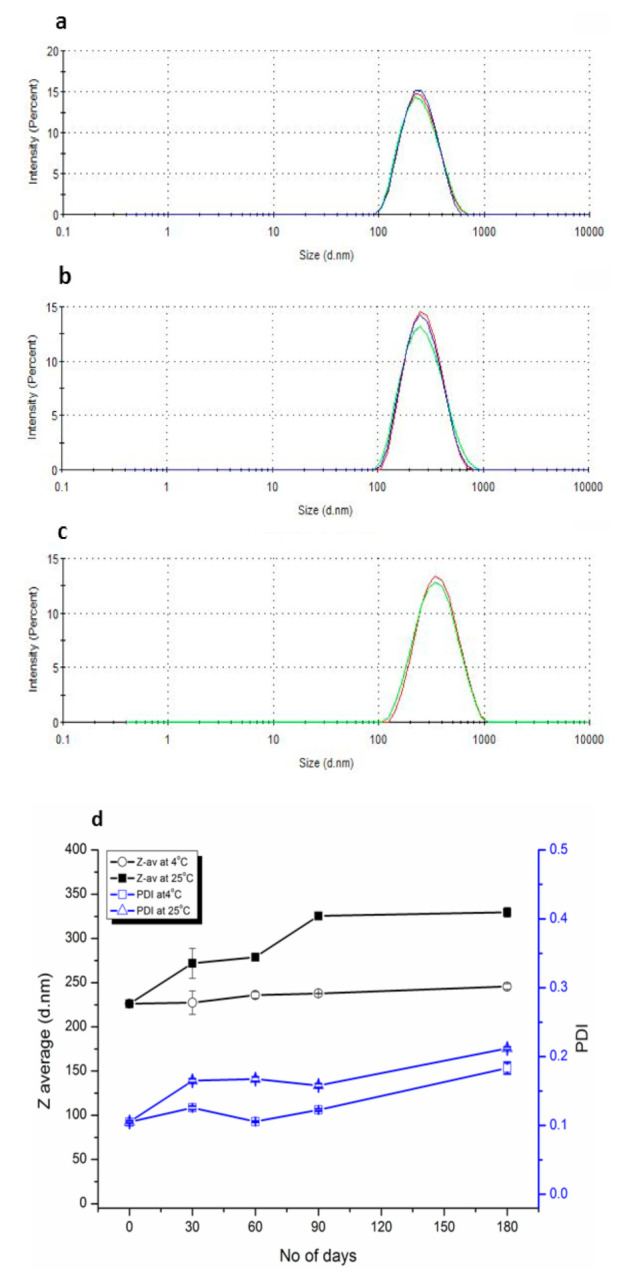
Accelerated stability study of optimized liposomes after sizing by serial membrane filtration with 0.2 µm membrane filter and stored in the primary hydration medium. (**a**): DLS results at 0-day. (**b**): DLS results after 180 days at 4 °C. (**c**): DLS results after 180 days at 25 °C. (**d**): Z-av and PDI vs. the number of days at 4 °C and 25 °C. The results are presented as mean ± SD.

**Table 1 pharmaceutics-12-01065-t001:** Key results of optimization studies through the CI method. The shortlisted results were highlighted and considered as a base for the next experiment. **a**: The quantity of organic-phase was varied against a fixed volume of aqueous-phase. **b**: The quantity of aqueous-phase was reduced against the pre-set quantity of organic-phase from step-a. **c**: The quantity of maximum Cho concentration against the pre-set quantity of HPCE was assessed with the optimized parameters of a and b. **d**: The minimum volume of organic-phase required for vesiculation was assessed with pre-sets a, b and c. **e**: The maximum concentration of amphiphiles in minimum solvent was assessed through variable temperature to evaluate the minimum possible temperature for vesiculation. ***** Size measurement was performed at standard dilution for a particular set of experiments.

Description	Process Variables					Results			
	HPCE/DW (µmol/mL)	Cho/DW (µmol/mL)	HPCE/CHCl_3_ (µmol/mL)	Cho/CHCl_3_ (µmol/mL)	Temp (°C)	Physical Appearance	Count Rate* (KCPs)	PDI	Z-av (d. nm)
a—Organic phase volume variables	0.15	0.20	1.50	2.00	25	Uniform	275.45 ± 3.42	0.22 ± 0.01	270 ± 17.30
	0.15	0.20	2.00	2.50	25	Uniform	276.57 ± 2.20	0.24 ± 0.01	225.70 ± 1.70
	0.15	0.20	3.00	3.75	25	Uniform	170.28 ± 2.00	0.24 ± 0.01	242.15 ± 1.45
	0.15	0.20	4.00	5.00	25	Uniform	157.70 ± 9.38	0.26 ± 0.01	271.65 ± 3.35
	**0.15**	**0.20**	**8.00**	**10.00**	**25**	**Uniform**	**148.39 ± 0.27**	**0.27 ± 0.01**	**285.80 ± 4.2**
b—Aqueous phase volume	0.19	0.25	8.00	10.00	25	Uniform	195.58 ± 0.36	0.14 ± 0.04	484.55 ± 28.95
	0.25	0.33	8.00	10.00	25	Uniform	402.68 ± 4.63	0.29 ± 0.00	264.65 ± 0.25
	**0.38**	**0.50**	**8.00**	**10.00**	**25**	**Uniform**	**746.52 ± 6.09**	**0.19 ± 0.02**	**243.20 ± 0.40**
	0.75	1.00	8.00	10.00	25	ppt	-	-	-
	1.50	2.00	8.00	10.00	25	ppt	-	-	-
	3.00	4.00	8.00	10.00	25	ppt	-	-	-
	6.00	8.00	8.00	10.00	25	ppt	-	-	-
c—Cho concentration variables	0.38	0.13	8.00	2.50	25	Uniform	227.50 ± 5.50	0.35 ± 0.01	1136.85 ± 34.85
	0.38	0.25	8.00	5.00	25	Uniform	342.65 ± 2.35	0.37 ± 0.02	1464.56 ± 71.26
	0.38	0.38	8.00	7.50	25	Uniform	175.15 ± 58.5	0.99 ± 0.01	3381.11 ± 5.98
	**0.38**	**0.50**	**8.00**	**10.00**	**25**	**Uniform**	**269.87 ± 5.03**	**0.26 ± 0.01**	**203.74 ± 5.94**
	0.38	0.65	8.00	12.50	25	ppt	-	-	-
	0.38	0.80	8.00	15.00	25	ppt	-	-	-
d—Minimum organic phase	0.38	0.50	10.00	15.00	25	Uniform	128.60 ± 1.90	0.39 ± 0.06	2917.3 ± 310.30
	**0.38**	**0.50**	**15.00**	**20.00**	**25**	**Uniform**	**122.50 ± 2.50**	**0.46 ± 0.03**	**2858.5 ± 438.87**
	**0.38**	**0.50**	**30.00**	**40.00**	**25**	**Uniform**	**116.50 ± 2.50**	**0.38 ± 0.03**	**3403.85 ± 178.55**
	0.38	0.50	60.00	80.00	25	ppt	-	-	-
e—Temperature variables	4.00	5.00	7.50	10.00	25	ppt	-	-	-
	4.00	5.00	15.00	20.00	25	ppt	-	-	-
	4.00	5.00	30.00	40.00	25	ppt	-	-	-
	4.00	5.00	7.50	10.00	35	ppt	-	-	-
	4.00	5.00	15.00	20.00	35	ppt	-	-	-
	4.00	5.00	30.00	40.00	35	ppt	-	-	-
	**4.00**	**5.00**	**7.50**	**10.00**	**45**	**ppt**	**262.53 ± 0.57**	**0.24 ± 0.06**	**188.35 ± 28.55**
	**4.00**	**5.00**	**15.00**	**20.00**	**45**	**Uniform**	**119.06 ± 1.74**	**0.42 ± 0.22**	**472.10 ± 19.9**
	**4.00**	**5.00**	**30.00**	**40.00**	**45**	**Uniform**	**157.42 ± 1.04**	**0.37 ± 0.07**	**786.85 ± 26.95**
	4.00	5.00	7.50	10.00	55	ppt	-	-	-
	4.00	5.00	15.00	20.00	55	Uniform	150.27 ± 1.10	0.91 ± 0.24	220.15 ± 64.85
	4.00	5.00	30.00	40.00	55	Uniform	140.50 ± 0.19	1.17 ± 0.18	274.00 ± 81.80

**Table 2 pharmaceutics-12-01065-t002:** Troubleshooting of expected processing problems while preparing liposomes with the CI and SPT methods.

Process problems	Rationale	Remedies and precautions
Solid depositions on the walls of the rotating flask.	(a) Lower processing temperature than required for lipids in use (optimized processing temperature is ≤10 °C below the transition temperature of the lipid). (b) The slow speed of the rotating flask. (c) The angle of the rotating flask exposing small surface area for evaporation.	(a) Continue mixing without vacuum. (b) Collect the uniform contents. Dissolve the solid deposits in fresh volume of chloroform. Add the collected portion to the above and proceed with rotary evaporation at the optimized parameters.
Encapsulation of thermolabile drugs.	Using high solids/solvent ratios, the methods are workable at ≤10 °C below the transition temperature of the lipid in use.	(a) Select a phospholipid with a lower transition temperature of ≤10 °C above the required process temperature. (b) Use the CI method with a low solid/solvent ratio. (c) Use large aqueous volumes.
Large values of Z-av and PDI.	(a) Filtration at a lower temperature than the processing temperature. (b) The difference in syringe to filter size ratio or sizing at lower pressure. (c) Electrical potential dependent size distribution.	(a) Processing temperature must be maintained during the sizing step. (b) Suitable syringe to filter disc size ratio must be maintained to build effective pressure for sizing. (c) ζ-potential must be considered and compressed for rational results.
Aggregation/precipitation after storage.	(a) Improper storage conditions. (b) Presence of unencapsulated material.	(a) Store at 5 ± 3 °C in light protected type-1 glass container. (b) Washout unencapsulated material through dialysis at 5 ± 3 °C. (c) Lyophilization.
Aggregation of solids at the base in the CI method.	(a) Lower temperature. (b) Vortex in the aqueous phase during the injection of amphiphiles. (c) Subsequent slow mixing neglected.	(a) Monitor the required temperature before injecting the organic phase in the CI method. (b) Avoid the vortexing of aqueous phase during the injection step. (c) Subsequent slow mixing for 2–3 min to obtain a uniform suspension. (d) Carefully transfer the aggregated material to the rotating flask. (e) Dissolve the aggregates in a fresh volume of chloroform and add to the rotating flask for further processing.

**Table 3 pharmaceutics-12-01065-t003:** Comparative study of optimized liposomes produced by the SPT method at a variable concentration of organic phase before and after sizing with serial membrane filtration.

HPCE/CHCl_3_		30.00 µmol/mL		15.00 µmol/mL		7.50 µmol/mL	
Sizing with 0.2 µm Filter		Unsized	Sized	Unsized	Sized	Unsized	Sized
Dynamic light scattering	Z-av in DW (d. nm)	2441.50 ± 15.50	243.80 ± 2.10	1824.50 ± 39.50	272.65 ± 1.95	1566.00 ± 236.00	257.45 ± 4.15
	PDI in DW	0.61 ± 0.19	0.10 ± 0.00	0.40 ± 0.01	0.15 ± 0.01	0.68 ± 0.09	0.10 ± 0.03
	Z-av in NaCl 10 µM (d. nm)	1466.50 ± 20.4	183.7 ± 1.90	928.05 ± 22.45	222.8 ± 2.10	1128.15 ± 95	214.7 ± 2.8
	PDI in NaCl 10 µM	0.320 ± 0.00	0.20 ± 0.00	0.400 ± 0.00	0.10 ± 0.00	0.340 ± 0.02	0.12 ± 0.02
	ζ-potential (−mV)	16.50 ± 0.3	17.10 ± 0.10	15.45 ± 0.35	16.60 ± 0.200	20.40 ± 0.6	18.15 ± 0.15
Atomic Force microscopy	Diameter (nm)	892.00 ± 98.06	152.00 ± 20.23	371.64 ± 87.18	149.25 ± 30.20	475.44 ± 165.36	179.17 ± 24.00
	Height (nm)	29.60 ± 8.85	24.57 ± 3.99	31.12 ± 4.64	30.58 ± 6.82	47.11 ± 20.48	23.83 ± 1.19
	Rigidity (h/d)	0.02 ± 0.01	0.13 ± 0.02	0.03 ± 0,01	0.14 ± 0.03	0.04 ± 0.02	0.11 ± 0.01
	Volume ×10^6^ (nm^3^)	19.2 ± 8.57	0.45 ± 0.17	3.73 ± 2.38	0.57 ± 0.39	11.52 ± 12.70	0.60 ± 0.20
Entrapped Volume (L/mol)		-	22.95 ± 0.07	-	23.87 ± 0.18	-	20.09 ± 0.18

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
