# Peer review of "Chloroform-Injection (CI) and Spontaneous-Phase-Transition (SPT) Are Novel Methods, Simplifying the Fabrication of Liposomes with Versatile Solution to Cholesterol Content and Size Distribution"

_pharmaceutics, 2020, doi:10.3390/pharmaceutics12111065_

Round 1

Reviewer 1 Report

The paper titled “Chloroform-injection (CI) and Spontaneous-phase-transition (SPT) are Novel Methods; Simplifying the Fabrication Liposomes with Versatile Solution to Cholesterol Contents and Size Distribution” compares the two methods and gives explanation for many problems and solutions. Even thou a lot of interesting information are given here; I have some marks and concerns.

Firstly, the paper is too extensive and it has to be more concise. The experimental design is not explained in the right modus. It is hard to follow the ideas and variables in the current form.

Further, the whole manuscript has to be upgraded in terms of English language. Many mistakes are found and some sentences don’t have right meaning or even don’t have a verb.

I also suggest to improve the introduction by excluding results and conclusions (in last paragraph) and including more convincing reason for usage and comparison of these two methods having in mind that solvent injection is not novel method. In current form the sense of the work is lost. It should be rewritten emphasizing both actual and potential uses of your research results.

The lipid composition of should be given

Please provide more information about high stirring during CI method (device, speed, time)

Please explain equation 3 in more details, it is a bit confusing or incorrect.

There is a sizing and purification section 2.3. but also sizing and purification in section 2.6. Whether all samples were treated in the same way? SUV are very unstable and inconvenient to structural studies

What diameter was determined for the liposomes?

Table 1, what was the difference in samples in part b and c when process variables are the same (0.38, 0.5, 8.00, 10.00, 25)? The results are drastically different

Table 1, part e: I am confused, what is the optimal HPCE/DW 0.38 or 4 and also Chol/DW 0.5 or 5, why you chose 4 and 5?

Table 2 is interesting but maybe as supplementary document.

Figure 1, please give the better quality and resolution of the graphs and pictures.

In Table 3 there are some Trial ID which are not used anywhere else in the manuscript.

Which was the criterion for say that liposomes are stable? The literature claims stable liposomes if their zeta values are higher than 20 mV

How did you determine the lamellarity?

The literature is quite old, please compare and discus your work with more recent literature.

What are the final lipid concentrations used here?

Reviewer 2 Report

The paper describes easy methods to make liposomes by mixing an organic phase containing phospholipids with double-distilled water. Optimal conditions are found to have high lipid/solvent ratio to have low amount of organic solvent to deal with and high entrapment power. Cholesterol helps in improving liposome stability and quality. I found the paper very stimolating and rich of information. The study is complete and give a wide picture of a many variables problem. 

However, I think it could be improved in some points: Tables of different results let me the idea that sample homogeneity, precipitation, aggregation is not very well controlled. In this sense I find difficult to accept that authors even in the best claimed conditions do not have problems with a) cholesterol solubility, b) control in the final liposome composition, c) real unilamellarity. They did not say the final lipid concentration after the whole process, and as a consequence if and how much mass they loose or have to recover, with respect the preset one. The phosphate determination seems to be done in some cases. When do they apply low rpm centrifugation to eliminate large particles or GUV? Not very clear to me.

Are the SEM images representative of the entire sample? The image resolution does not justify an unilamellar structure, it could well be multilamellar or a mixture, but to say that all the vesicles are LUV is a strong position, needing more strict evidences. The authors could leave open the question without concluding on this point. In fact further extrusion will increase particles unilamellarity,  but for example if small not unilamellar particles are present even the extrusion could not change much their structure.

In table 1 two different lines report the sampe kind of sample DPCE/DW=0.38 in b) and c) issues but different results. It is confusing to me.

The affirmation that the lower CR under filtration is a lost in vesicles is not correct, since by lowering mass particles by keeping the mass constant the light scattered decreases obviously, actually you can have even an increase in the particle number, with a quality change.

The formula 3 is wrongly written. I believe that the volume =V*Am/(M*As).

The expression is not clearly explained. It is not clear that the absorbance As comes from RhB dissolved in a volume with also triton, the same background composition present when liposomes are dissolved to measure the entrapped molecules. Moreover M is not a mole but the lipid molar concentration to be measured in the 3ml of volume chosen as the standard volume. 

Which phosphate assay is used is missing.

Minor

Code for the Lipid product, fatty acid composition and lipid transition temperature in the methods. 

KCP is not a known acronym. For instance I know Kcps (Kilo-count per second). So please just write down the meaning that is not count rate.

Which is the laser wavelength? This can help in understanding how the form factor of large particles does contribute to the scattered light intensity.

I would write room temperature instead of RT (it reminds something else)

and reference temperature instead of REF.

Line 290 DW:CHCl3=1:4 I think is an error.

ref 43 is not well cited. 

Round 2

Reviewer 1 Report

Manuscript "Chloroforminjection and Spontaneous-phase-transition (SPT) are Novel Methods; Simplifying the Fabrication of Liposomes with Versatile Solution to Cholesterol Content and Size Distribution" was upgraded according to reviewers comments. I still believe Table 2 should a supplementary
document.